# Role of the Hyaluronan Receptor, Stabilin-2/HARE, in Health and Disease

**DOI:** 10.3390/ijms21103504

**Published:** 2020-05-15

**Authors:** Edward N. Harris, Erika Baker

**Affiliations:** Department of Biochemistry, University of Nebraska, 1901 Vine St., Lincoln, NE 68588, USA; Erika.baker.35@gmail.com

**Keywords:** Stabilin-2, hyaluronan, HARE, MAPK, signaling, receptor, ligand, clearance

## Abstract

Stabilin-2/HARE is the primary clearance receptor for circulating hyaluronan (HA), a polysaccharide found in the extracellular matrix (ECM) of metazoans. HA has many biological functions including joint lubrication, ocular turgor pressure, skin elasticity and hydration, cell motility, and intercellular signaling, among many others. The regulatory system for HA content in the tissues, lymphatics, and circulatory systems is due, in part, to Stabilin-2/HARE. The activity of this receptor was discovered about 40 years ago (early 1980s), cloned in the mid-1990s, and has been characterized since then. Here, we discuss the overall domain organization of this receptor and how it correlates to ligand binding, cellular signaling, and its role in known physiological disorders such as cancer.

## 1. Introduction

The focus of this review is the interactions of hyaluronan (HA) with Stabilin-2 in both physiologically normal and diseased states. HA is a long unmodified polysaccharide containing alternating sugars consisting of glucuronic acid and N-acetylglucosamine. Stabilin-2 is one of two members of the scavenger receptor class H family and is the only one that binds HA. Although fifty-five percent of the amino acids that compose Stabilin-1 are homologous to those of Stabilin-2, it does not bind HA, and thus Stabilin-1 will not be reviewed in this manuscript. The biological activity of Stabilin-2 was first discovered by Fraser et al. in experiments in which they were calculating the clearance of HA in the blood of rabbits in the early 1980s [1,2]. Radioactive HA that is injected directly into an ear vein is sequestered in the liver. By 1983-4, there was good evidence that the sinusoidal endothelial cells of the liver contained a mechanism to specifically internalize HA [3,4]. The protein receptor was first purified from sinusoidal endothelial cell (SEC) membranes and antibodies were raised against an enriched HA-binding protein in 1997 by Yannariello-Brown et al. [5] followed by McCourt and coworkers in 1999 [6]. A partial protein sequence was identified and then cloned in rats. The novel receptor was named the hyaluronic acid receptor for endocytosis, or HARE [7]. Human HARE refers to the smaller 175-kDA isoform of a larger 300-kDa full-length receptor, which was entirely sequenced in 2002 [8]. At this time, it was named Stabilin-2, since the larger isoform had high homology and organization similar to Stabilin-1/MS-1. Around the same time in 2002, Stabilins also carried the name FEEL-1/FEEL-2, given that the proteins contained Fasciclin, epidermal growth factor-like (EGF), laminin-type EGF-like, and Link domains [9]. Since FEEL was a temporary name, it remains seldomly used to this day and Stabilin remains the widely accepted nomenclature of these receptors. The smaller isoform of Stabilin-2 is a proteolytically processed product of the larger isoform and has been independently characterized [10,11] (Figure 1). In this review, the smaller isoform is referred to as “HARE” and the larger isoform is Stabilin-2. To date, all of the biological functions of HARE and Stabilin-2 are identical, and both are expressed in tissues and recombinant cell lines.

## 2. Characteristics of Stabilin-2/HARE

As of this year (2020), there are 20 known ligands of Stabilin-2 and, despite this diversity, the protein receptor remains a mundane sequence of EGF and fasciclin domains. Stabilin-2 is composed of 2551 amino acids, of which 2458 are part of the extracellular domain, 21 are the single transmembrane domain, and the remaining 72 amino acids are the intracellular domain. There are 21 epidermal growth factor (EGF) and EGF-like domains in four clusters containing five to six domains per cluster (Figure 1). Each EGF domain has a specific folding pattern in which disulfide bonds form between the first and third, second and fourth, and fifth and sixth cysteines within that cluster [12]. However, the “defined” domain often has more than six cysteine residues within it. In between each EGF cluster, there are two fasciclin (Fas-1) domains often containing no cysteines. Only the fifth and sixth Fas1 domains contain a few cysteines. The seventh Fas1 domain is adjacent to the transmembrane domain and is the only one with a known structural organization [13]. Between the last EGF cluster and the seven Fas1 domain is the 93-amino acid Link domain, the known HA-binding domain, containing two disulfide bonds. Stabilin-2 contains 202 cysteines, of which 130 are known to be paired with each other in the EGF and Link domains. The oxidation fate of the remaining 72 cysteine residues is currently unknown. The following is what we know of the individual domains of Stabilin-2.
Fas1: Recombinant cells that express high levels of Stabilin-2 often become sticky in that they stick to each other. Park et al. investigated how this occurred and found that Fas1 domains are responsible for homophilic cell–cell adhesion which also required a divalent cation such as Mn^2+^, Mg^2+^ or Ca^2+^ [14]. Fas1 domains also interact with integrins in a heterophilic fashion. Using mouse recombinant L-cells expressing Stabilin-2 and following up with human SECs, they found that the Fas1 domains of Stabilin-2 adhere with αMβ2 integrins on the surface of leukocytes and that this adherence was firm and not associated with rolling nor transmigration [15]. Further studies by the same laboratory showed that aged or apoptotic cells such as erythrocytes are immobilized by Stabilin-2 through the interaction of αVβ5 integrins and that engulfment occurs via the GULP adaptor-mediated activation of Rac1. These experiments were also performed in L-cells stably expressing Stabilin-2 in which phagocytosis could occur [16]. Since SECs do not employ phagocytic mechanisms, SECs may immobilize the target cells in the sinusoid allowing Kupffer cells to engulf them. Alternatively, immune cells expressing Stabilin-2 may phagocytose dead or apoptotic cells.EGF: The inner leaflet of the plasma membrane of erythrocytes is enriched in phosphatidylserine (PS), which flips to the outer leaflet when the cells die [17]. The exposure of PS is a signal for phagocytic cells to “eat” the dead cell and clear it from circulation. Park et al. discovered that the EGF domains of Stabilin-2 (and Stabilin-1) recognize PS as a binding ligand and may immobilize apoptotic/aged cells in the sinusoids for engulfment by Kupffer cells [18,19]. The interaction required Ca^2+^ and occurred at pH 7.3, but was optimal at pH 6.8. Furthermore, the intriguing part of this story is that the second EGF domain in each cluster is more atypical than the other EGF domains and there is a conserved histidine which aligns with position 1403 of the third cluster. This histidine is conserved within each EGF cluster and between both Stabilin receptors. Mutagenesis of the histidine did not affect binding affinity for PS, but did affect binding enhancement at lower pH, suggesting that the histidine is protonated at lower pH (pKa ~6.0) and that other residues in the binding loop are involved with PS interaction [20].Link: The Link domain is the HA binding domain for Stabilin-2. It is located between the last EGF and Fas1 domains near the transmembrane region. Deletion of the Link domain results in complete ablation of HA binding [21,22]. The Link domain of Stabilin-2 has highest homology to the Link domain of tumor necrosis factor-stimulated gene-6 (TSG-6). In silico modeling suggests that the tyrosine residues are critical for HA binding [23]. Unlike the Stabilin-2 Link, the TSG-6 Link also binds heparin at a distal site within the domain [24]. Stabilin-2 also binds heparin, but not within the link domain and the precise location is unknown at this time [25]. Although HA binds the Stabilin-2 Link domain with high affinity approaching 20 nM, the interaction is relatively weak compared to other hyalectins. With the use of atomic force microscopy for direct measurement of the interaction of the protein with a low megaDalton polymer, HA bound to Stabilin-2 with 25 picoNewtons (pN) in comparison to TSG-6 (24 pN), CD44 (34 pN), versican (37 pN) and aggrecan (>52 pN) [26]. Having both high affinity and the ability to release the cargo once in the early endosomes is critical for this receptor, in contrast to the other hyalectins which only bind and form stable matrices.

As alluded to previously, HARE is a proteolytically processed form of Stabilin-2 in which the N-terminal 1135 amino acids are cleaved from the remaining 1416 amino acid receptor bound to the membrane at serine 1136. Recombinant cells expressing only HARE show great ligand binding and endocytic activity. Recombinant cells expressing the full-length Stabilin-2 cDNA express both Stabilin-2 and HARE in which the ratio of the two receptors is about 8-10:1 [10]. Expression of HARE is much more robust in the tissues of rodents (ratio ~2:1) and, presumably, humans, than in recombinant cell lines which may not fully express the proteolytic enzymes required for this processing event [27]. From ^35^S pulse-chase experiments, we know that HARE expression in Stabilin-2 expressing cells begins after the receptor is fully glycosylated; therefore, we have speculated that the cleavage event occurs somewhere between the trans-Golgi network and the cell surface [28]. We have used standard molecular biological techniques to delete the transmembrane and cytoplasmic domains from Stabilin-2 to produce a soluble “ecto-domain” [10]. Cells secreting the 300-kDa ecto-domain do not produce a soluble HARE, therefore, protein cleavage is dependent on membrane anchorage [29]. Secreted ecto-domains have proven quite useful to interrogate and calculate affinities for ligand binding using biochemical methods [10,22,30]. The ecto-domains are easily purified by affinity and metal chelate chromatography from cell culture media and they are quite stable under standard conditions [22]. Reducing the protein renders the protein unable to bind to any ligand, and therefore the secondary and tertiary structures of the receptor are vitally important for its activity.

## 3. Ligand Binding of the LINK Domain

The 93 amino acid Link domain of Stabilin-2 is 65% homologous and 46% identical with the Link domain of Stabilin-1. The cysteine residues are perfectly aligned, but the Stabilin-1 Link lacks the key lysine, tyrosine and additional residues that are predicted for HA binding based on the TSG-6 model [23,31] (Figure 2). Though it is known that Stabilin-2 binds in the low nM range for HA, there were questions as to what other glycosaminoglycans (GAGs) are bound within the same site. The rat HARE (rHARE) was initially cloned and expressed in SK-Hep cells in the laboratory of Prof. Paul Weigel and the Link domain was characterized by HA binding using competition assays with other potential ligands. Endocytosis of HA was competitive with the following chondroitin sulfates (CS) from strongest to weakest apparent affinities; CS-E, CS-A, CSD, CS-C, CS-B (Dermatan sulfate/DS). Other GAGs that did not bind in this assay were heparin, heparan sulfate, keratan sulfate, and non-sulfated chondroitin. Ligand binding was also temperature sensitive, as no ligands competed with HA at 4 °C [32].

Similar experiments were repeated for hHARE after it was stably expressed in Flp-In 293 cells. The chondroitin sulfate species with the highest to lowest apparent affinities in the endocytosis competition assay were: CS-A, (CS-C, CS-D, CS-E, and non-sulfated chondroitin were all about equal), and DS. Once again heparin, heparan sulfate, keratan sulfate did not compete. None of these GAGs competed for HA at 4 °C. Scatchard analysis of HA binding in hHARE cells reveals that there is a single class of non-interacting binding, suggesting that there is one receptor in these recombinant cells that binds HA. This analysis also revealed that the affinity or K_d_ value is 7.2 ± 1.2 nM and that there are about 118,000 binding sites or receptors on the cell surface [11].

With the development of using purified protein and individually labeled ligands, it was possible to screen without competing a known ligand such as HA. This overcomes the weaknesses of the competition assay such as (1) a weak ligand may not sufficiently compete with a strong ligand, (2) protein conformational change for one ligand may prevent the other ligand from binding, and (3) other non-competing binding sites will not be detected. To address this, a secreted ecto-domain of both human Stabilin-2 isoforms (s315 for the large Stabilin-2 receptor) and sHARE (for the smaller isoform) were created. Through a series of assays including ligand blots and an in-house development of an ELISA-like assay, the Link domain of sHARE binds HA, which was competed with by CS-A, CS-C, and CS-D. Both CS-E and DS bind to two sites and do compete slightly with HA bound in the Link domain. However, heparin binding was entirely independent of HA binding. This also confirmed that heparin does not bind within the Link domain and both ligands may bind sHARE simultaneously. CS-E and DS also compete with heparin at the second site outside of the Link domain [22].

## 4. Stabilin-2 as a Cellular Signaling Receptor

The cytoplasmic domain contains four Tyr, seven Ser, one His and five Thr amino acids, which are all potentially phosphorylated (Figure 3). Using NetPhos 2.0 software to predict phosphorylation probabilities, five candidates remain: Ser^2497^, Thr^2523^, Ser^2537^, Tyr^2519^, and Tyr^2531^. Of these, Tyr^2519^ was confirmed to be phosphorylated through a variety of biochemical and molecular approaches and interacts with the mitogen-activated protein kinase pathway (MAPK), specifically the extracellular signal-regulated kinases (ERK1/2) and not JNK or p38. The Ras-Raf-MEK-ERK cascade, or MAPK pathway, is involved with numerous cellular responses such as transcription, cell survival, migration, cytoskeletal remodeling, differentiation, and cell cycle progression in proliferation [33]. This pathway is often stimulated in many types of cancers and increased in roughly 1/3 of all cancers where multiple members of the cascade may be targeted. To test if Stabilin-2/HARE may be stimulated by ligand, we used HA due to its availability as a pure source from fermentation and because of its high affinity for the receptor. Incubation with a small amount of polymeric HA (40 nM) stimulated HEK293 cells stably expressing either Stabilin-2 or HARE to phosphorylate ERK1/2 which increased over a 30 min time course and then rapidly decreased [21]. This suggests that Stabilin-2 is responsive to extracellular clues and may signal that induction for a cellular response.

The cytoplasmic domains of both Stabilin receptors are very diverse, without any homology. As seen in Figure 3, the cytoplasmic domain of Stabilin-2 contains four endocytic motifs; YSYFRI, FQHF, NPLY, and DPL. Using deletion mutagenesis, the third motif, NPLY, was the most crucial for endocytosis. Deletions of either the first or second motif also inhibited endocytosis by 50%. The fourth motif decreased endocytosis by less than 10%. Swapping out Tyr^2519^ for alanine also significantly affected endocytosis, suggesting that receptor signaling upon ligand binding is a coordinated event [34]. We next asked if the type of ligand influences activation of some motifs over others. As seen in Figure 3, M1, M2, and M3 are activated by HA endocytosis, whereas, M1, M3, and M4 are activated for heparin endocytosis [35]. Since both HA and heparin are polymers that come in a range of sizes, HA is the most diverse with polymers containing only a few sugars to very long polymers up to 10 megaDaltons [36]. The current paradigm is that HA has multiple biological functions that are size-dependent in which smaller oligos are associated with inflammation and larger polymers tend to be in matrices or involved with cellular quiescence [37,38]. This paradigm is being put to the test due to sources of HA and methodology of small oligo production may introduce endotoxin in HA samples [39]. Using the HEK293 model system, it was discovered that HA stimulates the NF-kB pathway via the Link domain as cells expressing HARE without a Link domain were unresponsive. Using specifically sized HA preparations to probe this interaction proved that NF-kB is dependent on a range of 40–400 kDa HA. Polymers that were outside of this range failed to stimulate HARE-mediated NF-kB signaling [40]. Why would this matter? There are several hypotheses to this question. First, high molecular weight HA in the megaDalton size range is the normal mass of HA that can be found in certain tissues in which HA is found in an extracellular matrix bound by multiple proteins. HA is in a stable structure and there is no need for a response. Second, the 40–400 kDa size range may be optimal for receptor dimerization in contrast to small HAs that do not allow the bulk of two large receptors to attach efficiently or, alternatively, very large HA that does not allow the receptors to get close enough for any synergistic activity. Lastly, HA around the 200 kDa range may be the “sweet spot” for the transition from rod-like to coil-like HA [41]. HA in this size range signals through Stabilin-2/HARE for a cellular response, although this receptor may internalize any HA polymer greater than 18 sugars [40].

In an extension to the investigations of which ligands bound to HARE [22], we asked which of these activated NF-kB, or is HA the only relevant signaling molecule for Stabilin-2/HARE? Again, using stable cell lines, the inquiry led us to conclude that in addition to HA-mediated signaling, other ligands including heparin, DS, and acetylated low-density lipoprotein (AcLDL) also activated NF-kB. Other ligands that did not activate NF-kB were the chondroitin sulfates -A, -C, -D and -E [42]. It is worth noting that HA and the chondroitin sulfates are the only ligands to bind within the Link domain and with differences in their individual binding affinities. Heparin and DS bind to a distal site and compete with each other and with AcLDL, so that within this context, we are only looking at two binding sites on HARE: a HA/CS site which is within the Link domain and another site for Heparin, DS, CS-E, and AcLDL. In a collaborative effort with Anne Dell of the Imperial College of London starting in 2004, we screened the N-glycans of HARE and found a unique and large N-glycan that modified Asp^2280^ which is situated in the Link domain [28]. Of the 10 sites on HARE that were discovered to contain multiple N-glycans, only Asp^2280^ contained N-glycans that were sialylated. Mutagenesis of this Asp to Ala did not affect the endocytosis of HA, but did affect binding in some in vitro assays and we think that the absence of the glycan destabilized either the binding of HA with the protein or slightly altered the topology of the Link domain which lowered affinity in an in vitro experimental setting. With regard to signaling, the ablation of the glycan at position 2280 resulted in HA binding with the receptor without activation of NF-kB. In contrast, the N2280A mutant did not affect NF-kB signaling by heparin, DS, or AcLDL [43] suggesting that HA and the heparin cohort bind and induce signaling through the receptor independently. Due to the signaling activities of multiple ligands, including a size range of HA, we speculate that Stabilin-2/HARE is involved with sensing the extracellular environment via the profile or signature of the cohort of ligands in real time.

## 5. Physiological Functions of Stabilin-2

Stabilin-2 is expressed in the endothelium of liver [5], lymph node [27], spleen [27], bone marrow [44] and in specialized structures of the eye, heart, and kidney [45] (Table 1). The primary function of Stabilin-2 is dependent on the location and expression in the tissues. In the liver, it is highly expressed in liver sinusoidal endothelial cells (LSEC) and is responsible for the systemic clearance of HA [27], natural and infused heparin [25,46], extracellular matrix components such as chondroitin sulfates [32] and collagen pro-peptides [47], advanced glycation end-products [48] and small phosphatidylserine-based particles [49]. Stabilin-2 was first discovered for its HA binding ability. The circulatory route of HA catabolism, as established by T. Laurent and J.R. Fraser, is that very large megadalton-sized HA is broken down in the tissues and drains in the lymphatic system. High expression of Stabilin-2 in lymph nodes allows for at least 80% of the HA to be eliminated at that site. The remaining HA ends up in the blood and is taken up by LSECs, which keeps the amount of blood HA very low [50,51]. Deletion of StabilinStabilin-2 in a mouse knock-out model resulted in glomerular fibrosis which increased in severity when Stabilin-1 was also knocked out. The cause of the kidney damage was likely the lack of clearance of the TGF-β family member growth differentiation factor 15 (GDF-15), a ligand common for both StabilinStabilin-1 and Stabilin-2, and the accumulation of extracellular matrix material within the glomeruli [52]. The human body must also eliminate millions of dead red blood cells every second to achieve the turnover of all red blood cells within their lifespan of ~120 days. Stabilin-2 (and StabilinStabilin-1) may also adhere to dying and dead red blood cells and neutrophils, which have flipped phosphatidylserine to their outer membrane, in the capillary bed of the liver. The shear stress of circulating blood is very low in these areas of the liver and the StabilinStabilins may immobilize these dead/dying cells for engulfment by Kupffer cells [19,53,54].

Recently, it was discovered in specialized structures of the muscle [55]. In the muscle, Stabilin-2 is involved with myoblast fusion during muscle formation and regeneration, in which it interacts with phosphatidylserine [56]. Embryonic muscle formation or repair involves myogenic precursors that mature to myoblasts which fuse with each other to give rise to a multi-nucleated myotube. The myotube further matures to form skeletal muscle tissue [57]. Previous screenings of Stabilin-2 expressing tissues missed muscle since Stabilin-2 is only expressed during myogenic differentiation through calcineurin signaling [55]. Follow-up studies investigated how this sequence of events occurred from PS binding to cellular response. Hamoud et al. demonstrated that Stabilin-2 binding with PS stimulated the G-protein coupled receptor (GPCR) activity of BAI3 during myoblast fusion or in efferocytosis (phagocytosis of apoptotic bodies). BAI1-3 are adhesion GPCRs containing thrombspondin repeats (TSRs) on the extracellular tail and an ELMO binding site on the intracellular tail [58]. BAI3 GPCR activity was carried out through the ELMO-DOCK1-Rac1 pathway to rearrange the cytoskeleton and complete myotube formation or completion of the efferocytosis phagosome [59]. In this process, it is likely that the extracellular Fas1 domain of Stabilin-2 interacts with the β5 integrin domain of focal adhesion kinase and GULP, which is an adaptor protein for endocytosis and possibly connected to the activities of Rac1 for actin rearrangement, all aiding in the stimulation and formation of the cupping processes of these two events [16,60,61]. In addition, extracellular signaling for changes in transcription during myoblast fusion and myotube formation are heavily reliant on the MAPK pathway. As alluded to previously, HA binding to Stabilin-2 induces the MAPK pathway for inducing cellular changes and response [57]. For more information, Park and Kim have written an excellent review delineating these processes in the journal *Biomolecules* [49].

Both Stabilins are expressed in bone marrow and are involved with the clearance of HA and other extracellular matrix molecules in that localized area. Within the marrow, the endothelia is a migration site for hematopoietic stem and progenitor cells (HSPCs) between the blood and bone [62]. Do the Stabilins have a role in cell migration in and out of the bone marrow? This question was tested by the transfection of HEK293 cells with either Stabilin-1 or Stabilin-2 and assessing the binding of HSPCs to these transfected cells. Stabilin-2 expressing HEK293 cells bind bone marrow cells with higher affinity which is abrogated with the treatment of hyaluronidase [44]. Based on these experimental data, it is highly probable that the Stabilins, particularly Stabilin-2/HARE, with its HA-binding abilities, are involved with transmigration of HSPCs between bone marrow and circulating blood.

The human protein atlas (www.proteinatlas.org) and other similar databases using RNA expression data show that Stabilin-2 expression is highest in the spleen. A key word search in PubMed using “Stabilin-2 and spleen” brings up 11 references, none of which are a detailed study of Stabilin-2 function in spleen. A similar search in the Web of Science database by Clarivate Analytics produces similar results. What is the function of Stabilin-2 in spleen? We assume that it may be similar to bone marrow in that there is local clearance by the extracellular matrix and other functions that are specialized in the spleen, such as the enhanced elimination of dead blood cells, and, possibly, elimination of bacteria [9]. We should note that a small study using human cDNA pools from the spleen as well as from the lymph node and bone marrow identified splice variants in these tissues. Of the nine splice variants identified, six were in the spleen and the other three were in the lymph node and bone marrow. As the identification was based on RNA expression, it is unknown if these variants are expressed on the protein level nor their significance to human biology [63]. A more recent paper published in the J. of Clinical Investigations exhibits how Stabilin-2 is a clearance receptor for von Willibrand Factor (VWF) and Factor VIII (FVIII) of the coagulation pathway [64]. Both VWF and FVIII are naturally conjugated together in the plasma and levels of these molecules are regulated by Stabilin-2 clearance/endocytosis activity. The rate of VWF-FVIII clearance was only modestly decreased in a Stabilin-2 knock-out (Stab2KO) model, suggesting there are other clearance receptors that also recognize the VWF-FVIII complex [65]. Interestingly, HA, along with unfractionated heparin, dermatan sulfate and mannan, competed with VWF-FVIII binding to Stabilin-2 and the presence of HA reduced the titer of FVIII-specific IgGs in a similar manner as observed in the Stab2KO background [64]. It is thought that the mechanism for the immune response to FVIII is in the spleen, the site of very high Stabilin-2 expression, though how Stabilin-2 affects overall titers of FVIII-specific IgGs and immunotolerance is unknown [66]. This may very well be a case in which levels of VWF-FVIII are regulated by Stabilin-2 in the liver and the immune component may be regulated by Stabilin-2 in the spleen. To date, there is not one detailed study of Stabilin-2 function in the spleen and it is an area ripe for exploration.

## 6. Stabilin-2 and Cancer Metastasis

In 1889, Dr. Stephen Paget’s “seed and soil” theory of metastasis stated that a tumor cell (or seed) will find a home in certain compatible tissues (soil) to continue growing in a suitable environment [68]. Tissues with the highest expression of Stabilin-2 (sinusoids of liver, lymph node) are also the primary targets (soil) of metastatic cancers from breast, colon, prostate, gastrointestinal, etc. (seeds) [69]. Metastatic tumor cells often contain distinctive surface markers for enabling them to escape their indigenous tissue to travel and thrive at a distal site. One of the common markers is CD44 and its splice variants and this is why it has been of such interest in cancer research. CD44 is a HA binding molecule which is not highly endocytic and many cancers (as well as immune cells) express variants of CD44 and have abundant pericellular HA [70,71]. HA is a ubiquitous molecule and it actually helps to “cloak” or “hide” the cancer cell from immune surveillance. Metastatic cells that are not detected by the immune system often immobilize in the sinusoids of liver, lymph node and bone marrow. To investigate why this occurs, Martens et al. analyzed the scavenger receptor profile of these tissues and found a number of receptors that are pattern recognition receptors as well as specific ligand receptors. These include both Stabilin-1 and -2, DC-SIGN, mannose receptor (MR), MARCO, and LYVE-1 expressed by the endothelium and resident macrophages expressing MR, DC-SIGN, Sialoadhesin, CD163, and Thrombomodulin. These receptors represent a trapping mechanism for metastatic cells. Of these 11 receptors, only Stabilin-2 recognizes and binds with HA. It is interesting to note that the organ with the highest expression of Stabilin-2 is the spleen, though it is not a site for tumor metastasis. The spleen only expressed four of the 11 receptors listed here and so it may not have the optimal profile to immobilize metastatic cells [72]. In light of the hypothesis that Stabilin-2 may aid in the retention of metastatic cells, a study focused on the remodeling of endothelium in hepatocellular carcinoma (HCC) revealed that the loss of Stabilin-2 expression increased survival of the patients that were sampled. As normal cells become cancerous, they de-differentiate and lose many of their tissue-specific markers. Unfortunately, loss of Stabilin-2 expression in the peri-tumorous environment was the least likely to occur of all the SEC markers tested and may be a significant factor for endothelial-tumor cell adhesion and invasion [73].

The lymphatic system or “second circulatory system” is a common transportation highway for metastases and tumor lymphangiogenesis and is common among the various types of cancer. The lymph node itself highly expresses Stabilin-2 and acts as one of the entrapping receptors for metastatic tumor cells. A study in which the investigators injected PC3M-LN4 prostate cancer cells that are known to be rich in pericellular HA in mice, found that when Stabilin-2 was blocked with a blocking antibody prior to tumor injection, the number of lymph node metastases was dramatically reduced [74]. A follow-up human patient study in the evaluation of tongue cancer supports this finding in which the potential of solid tumor lymph node metastases is positively correlated with Stabilin-2 expression in lymph node [75]. Though it is known that pericellular HA increases metastatic potential, it also increases interaction with Stabilin-2 which allows for tumors to grow at distal sites.

Stabilin-2 is the primary scavenger receptor for HA as mentioned previously. Mice lacking Stabilin-2 expression (stab2KO) have very high HA levels in their circulating plasma, though the mice are physiologically normal in all other aspects [52]. The role of circulating HA in a metastatic model was carried out in stab2KO mice in which mice were injected with B16F10 melanoma cells and the number of metastatic nodules was assessed in the lungs. The results are that (1) the number of metastatic nodules was lower in Stab2KO than WT mice, (2) there was significantly less rolling and tethering of cells in the Stab2KO mice compared to WT, suggesting the importance of circulating HA, or rather the lack of it, promotes metastatic cellular attachment [76]. This paper demonstrates that Stabilin-2 does not need to directly interact with metastatic cells, but the levels of ligands also affect metastatic activity.

CD44 and its splice variants are enriched in a variety of cancer cells and all of these receptors bind HA [77,78]. Therefore, the use of HA as a vehicle for nanoparticles to target cancer cells may be a promising therapeutic tool. Evaluation of HA-based nanoparticles often begin and end in the cell culture stage without further assessment in a physiological model [79]. In the experimental animal, nanoparticles coated with HA often accumulate in the liver due to Stabilin-2 binding and endocytosis activities [80,81]. If HA is required for nanoparticle targeting, then it must be physically or chemically modified to optimize delivery to the target tissue while avoiding accumulation within the liver. For example, carboxymethylation of HA allowed for greater drug sequestration and overall efficacy of drug delivery within a limited time window, though the study lacked some key data for liver or renal clearance [82]. Likewise, nanoparticles coated with low molecular weight HA have higher internalization rates by tumor cells and better efficacy than nanoparticles without HA and little to no toxicity to either liver or kidney. However, other treatments, such as radiation, were required to significantly hinder growth of tumors [83]. The bottom line is that many nanoparticles have been developed for drug delivery to a variety different types of cancer and there is not yet a developed HA-based nanoparticle that contains optimal drug delivery to the target tissue without off-target effects while retaining the intrinsic properties of HA for binding to key receptors [84,85,86].

## 7. Conclusions and Perspectives

By definition, Stabilin-2/HARE is an excellent scavenger receptor for the following reasons: (1) it binds and internalizes both natural and synthetic ligands, (2) it continuously recycles from the cell surface to endosomes to accumulate exogenous cargo, and (3) the expression is relatively high in specific cell types of tissues and organs. Moreover, the function of Stabilin-2/HARE is ever expanding to include aspects of development. The recent discovery that Stabilin-2 is involved with myotube development was briefly covered [55]. There are other reports that Stabilin-2 is involved with development in which expression oscillates in the fetal liver (unpublished data) or involved with arterial–venous differentiation in the zebrafish model [87]. What is known now is only scratching the surface of the myriad functions for Stabilin-2 in tissue/organ development.

One of the big impediments of characterizing this receptor on the atomic level is the lack of a known structure. Both isoforms contain an abundance of cysteine amino acids which are difficult to fold in bacterial-based expression systems. Dr. In-San Kim’s lab used an Origami strain of *E. coli* to express his constructs in Park et al. [56] and correct protein folding was estimated at 70%. More recently, Twarda-Clapa et al. used an *E. coli* K12 Shuffle T7 Express Stain to express the seventh Fas-1 domain for crystallization, though this is one of the few areas of the protein without any cysteines [13]. Their resolution was down to 0.97 Angstrom and the folding is more complex than anticipated as predicted by the primary sequence. Glycosylation is another factor to consider for crystallization and we have found 10 sites within HARE and there are likely more in Stabilin-2. Structurally identifying individual domains may be misleading due to the number of unknown disulfide bonds in a folded and functional receptor. The best candidate for crystallizing this protein for understanding its overall structure is probably the soluble form of HARE expressed in either mammalian or insect cell systems which will fold it with high fidelity. Understanding how Stabilin-2 interacts with its ligands and functions with other proteins will require structural modeling going forward.

The hepatic and renal systems eliminate nearly all of the foreign molecules that are either injected or consumed. Very small molecules tend to be eliminated through the kidney, whereas larger molecules or more complex structures such as polymers or nanoparticles are cleared through the liver [88,89]. Within the liver, injected pharmaceuticals or material absorbed from oral ingestion into the circulatory system is cleared by receptor-mediated LSEC scavenging or by hepatobiliary elimination. Due to the prolific endocytic activity of LSECs and, to a lesser extent, other liver cells, pharmaceuticals or their carriers may either target or avoid Stabilins or other scavenger receptors of the liver. Our work using phosphorothioate-based antisense oligonucleotides (PS-ASOs), which are used to downregulate gene expression or correct RNA splicing for human therapy, indicates that PS-ASOs that are highly negatively charged and with a flexible structure (single stranded) will bind with the Stabilins, whereas more neutral chemistries or double stranded ASOs will not [30,90]. Likewise, ammonium-functionalized carbon nanotubes (fCNTs), which are anionic and considerably larger, bind specifically with the Stabilins and are taken up by LSECs and not by Kupffer cells nor hepatocytes [91]. Studies in the zebrafish model suggested that small anionic nanoparticles that are 50-250 nm in size tend to be taken up by the Stabilins [92]. This fits within the theoretical limit of forming pinocytic vesicles which are, on average, 100–130 nm in diameter. Altering the shape of the nanoparticle and the chemistry for overall charge will affect its affinity for the Stabilins, in particular, Stabilin-2 [93]. With that in mind, the design of pharmaceuticals and their carriers need to be specific for either targeting or avoiding the Stabilins for optimal efficacy.

## Figures and Tables

**Figure 1 ijms-21-03504-f001:**
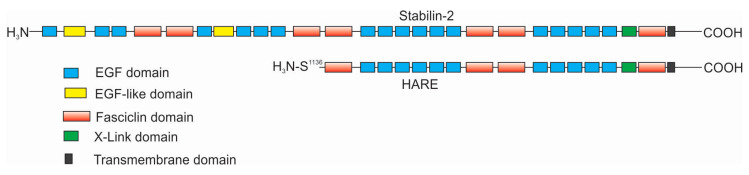
Domain organization of Stabilin-2 and the shorter isoform, hyaluronic acid receptor for endocytosis (HARE).

**Figure 2 ijms-21-03504-f002:**
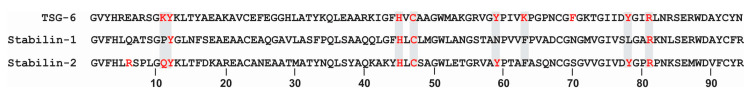
Alignment of Link domains in TSG-6, Stab1 and Stab2.

**Figure 3 ijms-21-03504-f003:**
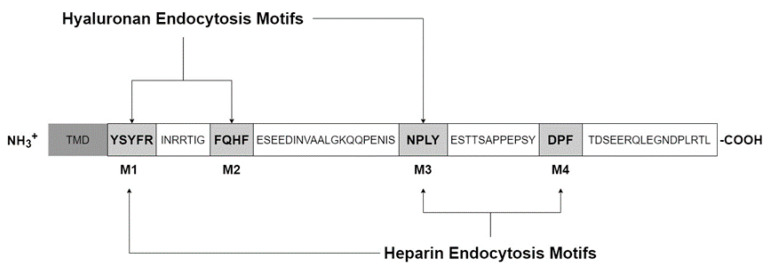
The cytoplasmic domain of Stabilin-2/HARE.

**Table 1 ijms-21-03504-t001:** Expression and Function of Stabilin-2/HARE.

Organ	Tissue	Function	Reference
Liver	Sinusoidal endothelia	HA and other blood borne molecule catabolism	[8,45,67]
Lymph node	Medullary sinuses	HA catabolism	[8,45,67]
Spleen	Venous sinusoids	Not Determined	[8,45,67]
Bone marrow	Venous sinusoids	Bone cell homing	[44,45]
Eye	Corneal and cuboidal epithelium	Not Determined	[45]
Kidney	Renal papillae	Not Determined	[45]
Brain	Ependymal cells of choroid plexuses	Not Determined	[45]
Heart	Mesenchymal cells of heart valves	Not Determined	[45]
Striated Muscle	Myocyte	Myoblast fusion in myogenesis	[55,59]
Blood	Human monocyte derived macrophages	Clearance of cell corpse	[18]

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
