# Peer review of "Role of the Hyaluronan Receptor, Stabilin-2/HARE, in Health and Disease"

_ijms, 2020, doi:10.3390/ijms21103504_

Round 1

Reviewer 1 Report

This is a very well written and interesting review by experts in the field covering the structure, ligand binding and cell signalling properties of stabilin-2.  

I have some additional comments

The review gives an in depth overview of the receptor function of in terms of ligand binding and cellular signalling but the title of the review is the role of stabilin-1 in health and disease. With this in mind, my suggestion would be that the article needs further additional detail with an extra named section from line 227 onwards- eg ‘ The expression and physiological functions of stabilin-2 in health’ ? -   detailing the expression of stabilin-1 during embryological development,  expanding on its role in HA uptake, apoptotic cell recognition and myocyte fusion.    Whilst this article is focused on stabilin-2-  I would recommend highlighting the consequences of knockout of the stabilin family which leads to renal fibrosis (Schledzewski K et al JCI 121 (2011) 703-12))  demonstrating the important role that stabilins play in maintaining homeostasis.  I would also include the 2018 paper from Swystun LL et al (JCI 2018 Aug 31;128(9):4057-4073) which explores the role of stabilin-2 in regulating the levels of von Willebrand factor-factor VIII and the link between clearance receptors and immunoregulation. 

The article would also benefit from a stabilin-2 overview figure - summarising the expression of stabilin-2 in the body  including various endothelial sites and myocytes and the potential roles it has in  homeostasis and disease. 

The conclusion section would also benefit from the authors expanding on therapeutic opportunities of targeting stabilin-2 and where future studies could help in the design of new precision medicine treatments and clinical translation  e.g. exploiting the scavenger function of stabilin-2 to take up nano-particles as a way of targeting liver sinusoidal endothelial cells  (Campbell et al ACS Nano 2018 Mar 27;12(13):218-2150) and cancer therapeutics. 

Author Response

We thank the reviewers for their efforts in improving this review article.  Revised and added text is outlined in red.

R1:  Whilst this article is focused on stabilin-2-  I would recommend highlighting the consequences of knockout of the stabilin family which leads to renal fibrosis (Schledzewski K et al JCI 121 (2011) 703-12))  demonstrating the important role that stabilins play in maintaining homeostasis.  I would also include the 2018 paper from Swystun LL et al (JCI 2018 Aug 31;128(9):4057-4073) which explores the role of stabilin-2 in regulating the levels of von Willebrand factor-factor VIII and the link between clearance receptors and immunoregulation.

Response:  We had an internal debate about how much more in-depth we should go with this article and we should have gone deeper.  We have added several paragraphs within the existing text and added a new section as advised.

R1: The article would also benefit from a stabilin-2 overview figure - summarising the expression of stabilin-2 in the body  including various endothelial sites and myocytes and the potential roles it has in  homeostasis and disease. 

Response:  We have added a table in section 5 to address this.

R1:  The conclusion section would also benefit from the authors expanding on therapeutic opportunities of targeting stabilin-2 and where future studies could help in the design of new precision medicine treatments and clinical translation 

Response:  We have added additional language addressing this.

Reviewer 2 Report

The manuscript is well written, containing relevant novel advancements within the studied subject and appropriate and relevant literature references. There are present just few minor points, which decrease overall quality of the manuscript:

  1. Figs. 1-3 are not of enough adequate graphic quality. 
  2. Line 136 and 141: "4° C" has to be corrected to "4 °C".
  3. Lines 248-249: This sentence has to be completely rewritten. There should be pointed: what were the main findings and why they are important.
  4. Line 327 as well as 329: "E. coli" to "E. coli"
  5. The references in the list are not formated according to the instructions for authors.

All in all, the review paper is well written and stuctured. Only minor text improvements are needed.

Author Response

We thank the reviewers for their efforts in improving this review article.  Revised and added text is outlined in red.

R2: Minor points.

  1. 1-3 are not of enough adequate graphic quality. 
  2. Line 136 and 141: "4° C" has to be corrected to "4 °C".
  3. Lines 248-249: This sentence has to be completely rewritten. There should be pointed: what were the main findings and why they are important.
  4. Line 327 as well as 329: "E. coli" to " coli"
  5. The references in the list are not formated according to the instructions for authors.

Response:  These points were all addressed in the revision.  We used Corel Draw professional software to produce the figures.